# A High Time-Efficient Missing Tag Detection Scheme for Integrated RFID Systems

**DOI:** 10.3390/s22124601

**Published:** 2022-06-18

**Authors:** Kaimin Guo, Xin Xie, Heng Qi, Keqiu Li

**Affiliations:** 1School of Computer Science and Technology, Dalian University of Technology, Dalian 116024, China; qhclement@gmail.com (H.Q.); keqiu@dlut.edu.cn (K.L.); 2Department of Computing, The Hong Kong Polytechnic University, Hong Kong, China; xiexindut@gmail.com

**Keywords:** RFID, multiple categories, multi-reader, detection of missing tags

## Abstract

Missing tag incidents are common in RFID-enabled supply-chain and warehousing scenarios due to cargo theft and employee error operations, which may lead to serious economic losses or potential safety hazards. On the premise of ensuring the accuracy of missing tag detection, this paper aims to improve the time efficiency in an integrated RFID system. Unlike prior work focusing on detecting missing items from a large number of homogeneous tags that are monitored by a single reader, one integrated RFID system possesses multiple readers to communicate with the heterogeneous tags, which have different categorical attributes. In addition, the prior work required repeating the execution several times to capture the missing tags in assorted categories, which is of low time efficiency. Thus, a protocol called Multi-reader Missing Tag Detection (MMTD) is proposed to capture the missing tag quickly and reliably, which can detect missing tags from different categories in a parallel manner and is much more time-efficient than previous work. MMTD has two major advantages compared to prior work: (i) It leverages the knowledge of the spatial distribution of tags to divide up a difficult detection task into several lightweight tasks, which are shared by multiple readers. (ii) It personalizes the time frame of the reader based on the tag population to optimize the utilization of the communication channel. The final simulation results reveal that MMTD is the best in time-efficiency among the comparison protocols, and MMTD outperforms the other missing tag detection protocols by at least 1.5× in the Integrated RFID scenarios.

## 1. Introduction

Radio Frequency Identification (RFID) technology has helped shape many aspects of our daily lives. For example, the items are attached with tags to quickly inventory in a warehouse and pay conveniently in an unattended supermarket. Missing tag detection is a valuable and important problem in RFID research. According to a recent survey [1,2], the American retail industry has lost $46.8 billion from the missing items in 2017. To avoid significant economic loss, the system manager wants to detect missing items as quickly as possible. Thus, time-efficiency is a critical metric in the evaluation of execution performance.

The related protocols can be classified into two groups: missing tag detection [3,4,5,6,7,8], and missing tag identification [9,10,11,12]. The former concentrates on detecting the number of missing tags with the principle of probability and statistics. Once a missing tag is detected, the reader will terminate the time frame immediately and raise a warning message [13,14]. The latter focuses on continuously obtaining the exact missing tags list. It will check the existence of each tag one by one in a deterministic manner. Compared with missing tag detection, the latter typically takes more time to obtain the exact ID of each missing tag. To balance accuracy and time-efficiency, the above two schemes are often integrated to be used simultaneously. Specifically, we can first apply a lightweight probabilistic tag detection scheme to continuously monitor the missing tags. Once a missing tag is found, we can further adopt the deterministic tag identification scheme to obtain the complete ID.

This paper aims to improve the time-efficiency of missing tag detection in integrated RFID systems. Although many protocols have been proposed, they simply apply a single reader to monitor one category of tags [3,7,15]. However, the reading range of a reader (even equipped with multiple antennas) is limited [16]. Therefore, in integrated RFID systems, many readers are deployed to ensure communication between readers and all tags. Thus, tags in the same category may be monitored by several readers and each reader covering parts of several categories. Some authors take the multi-reader and multi-category scenarios into consideration, but ignore the distribution of the tags and just set the uniform parameters for all readers [6]. Due to the differentiated spatial distribution of the tags, the uniform reader settings will result in more invalid slots and decrease detection efficiency. Therefore, personalizing the parameter settings of the reader is extremely important for improving the execution speed of missing tag detection.

## 2. Proposed Approach

To improve the time-efficiency in complex multi-reader and multi-category RFID scenarios, we propose a Multi-reader Missing Tag Detection (MMTD) protocol to parallelize the non-conflict readers and personalize the parameters of readers according to the tag population. In an integrated RFID system, readers are distributed to continuously investigate the monitoring area and determine if any of the tags are missing. However, reader-to-reader collision (R2Rc) happens when two readers have a duplicate communication area. R2Rc may result in identification failure for the tags, and many anti-conflict protocols are proposed to eliminate the effect of R2Rc [17,18]. In our simulation, we employ the greedy algorithm to group the tags into several isolation batches, which ensures that the tags in the same batch are free of collisions. As shown in Figure 1, the reader will be grouped into two batches labeled with black and red. During the tag querying process, we will activate readers according to their batch ordering.

MMTD can efficiently detect missing tags through the following three phases. First, each reader estimates the category size in its monitoring region and groups the tag categories with similar sizes for parallel missing tag detection. Second, since the categories in one group have similar sizes, the reader can adopt uniform parameter settings to detect missing tags in its region simultaneously. The reader repeats the above operation multiple times to detect missing tags in different groups. Finally, each reader can obtain a set of absent tags for each category, which contains both the extraterritorial tags and parts of the missing tags. The intersections of the absent sets are truly missing tags detected in the RFID system. For the first and second phases we can consult the paper [6]. However, this paper addresses the missing tag detection under the precondition that only one reader in the RFID system is used. We concentrate on the individuation of readers, here, we personalize the time frame of readers based on the tag population to optimize the utilization of the communication channel. Previous work commonly assumed that multiple readers operate simultaneously and execute the same parameters; therefore, these protocols are low detection accuracy and inefficient, due to the identification failure being led by R2Rc and these protocols do not take the differences between the tasks undertaken by readers into consideration. In MMTD, a difficult detection task is divided up and shared with the readers. The main challenges of this paper are demonstrated in the following content. The first challenge is to ensure sufficient detection accuracy for each category of tags $C_{i}$ and optimize the time-efficiency of MMTD’s. The second challenge is to dynamically adjust and determine the termination of MMTD to accommodate the diversity of category sizes in integrated RFID systems. Therefore, we summarize the main contributions of MMTD as follows: first, MMTD personalizes the parameters of readers to achieve higher execution speed in detection of missing tag; second, MMTD optimizes the broadcast frame size of each reader with sufficient theoretical analysis to balance both accuracy and time-efficiency; finally, we evaluate the performance of our MMTD through the extensive comparisons, and the results indicate that MMTD meets the required detection accuracy and that the execution speed is at least 1.5× faster than the other related protocols.

The rest of this paper is organized as follows. Section 3 elaborates on the advantages and disadvantages of related work on missing tag detection. Section 4 presents the system model and problem definition. Section 5 describes MMTD in detail and optimizes the parameters of readers. Section 6 conducts extensive simulation and evaluates the performance of MMTD. Section 7 is the conclusion of this paper.

## 3. Relate Work

RFID researchers have made great contributions to RFID development, and these works include the estimation of RFID tags [19,20,21], missing tag detection [3,6,7,8,22,23] and missing tags identification [9,10,12].

### 3.1. Estimation of RFID Tags

The estimation of RFID tags is to obtain a non-absolute value of the number of tags with the required accuracy. Zero-One Estimator (ZOE) [19] can achieve an accurate value of the number of tags with a high time-efficiency. In ZOE, only a small portion of tags are demanded to respond to the reader’s inquiry, and the cardinality of tags is inferred by measuring the ratio of busy (idle) slots. However, in an integrated RFID system, ZOE has to operate the estimations category-by-category. The Simultaneous Estimation for Multi-category RFID system (SEM) [20] exploits a new decoding mechanism that separates the single tag reply message from the combined signal in one slot, and estimates the category size by calculating the ratio of empty slots in the time frame. The RFID Estimation scheme with Blocker tags (REB) is the first estimation scheme that is designed for the privacy protection, REB imports the blocker tags, and calculates the population size of genuine tags by computing the ratio of empty slots and singleton slots in a time frame. Physical Layer Cardinality Estimation for large-scale RFID system (PLACE) [21] can obtain the number of tags in one slots through the signal of physical layer, and estimate the population size of tags by calculating the ratio of various collision slots. However, PLACE is designed based on the GNURario/USRP platform. The physical layer signals are difficult to achieve with commercial RFID systems. In this paper, we adopt the SEM method to estimate the population size of tags. SEM is designed for multi-category RFID system, and the simulations verify that it has excellent performance.

### 3.2. Missing Tag Detection

In the Efficient Missing-tag Detection protocol (EMD) [8] that Luo et al. proposed, the sampling method allows for a small ratio of tags participating in polling ID from reader to tags. In the Trusted Reader Protocol (TRP) [7], the frame slots are optimized and the missing tags can be detected by comparing the expected slots with actual slots. In the RFID monitoring protocol with unexpected tags (Run) [3] that Shahzad et al. proposed, multiple Aloha frames are used to detect the missing tag with unexpected tags existing. The Multi-seed Missing tag detection protocol (MSMD) [22] adopts the sampling and multi-hash methods to select the singleton slots in frames, if a singleton slot is transformed into empty, the missing tag event is existing. They all assume that one tag category and a single reader in work scenarios. Improving the time-efficiency and satisfying the accuracy of requirements are critical demands for an integrated RFID system. Liu et al. proposed the Simultaneous Missing Tag Detection protocol (SMTD) [6] for multiple categories RFID systems. SMTD decodes the combined signal of tag responses to detect if they are missing tags. However, in an integrated RFID system, where the readers of SMTD share uniform parameters, the distribution of tags and the individuation of readers are neglected.

### 3.3. Missing Tags Identification

Zhang et al. [10] uses the empty slots, singleton slots, that the particular tags should map into, to find missing tags. Yu et al. proposed the point-to-multipoint protocol (P2M) [9], in P2M, each tag chooses one slot to map, and the number of slots is $2^{Q}-1$. By optimizing the Q, P2M can achieve the minimum time-cost of missing tag identification. Vector-based Missing Key tag Identification protocol (VEKI) [12] needs to be executed multiple rounds, and the reader broadcasts an expected vector that is computed by the back-end server to tags. The missing tag will be determined by comparing the status of slots between an expected vector and actual frame slots.

## 4. System Model and Problem Definition

### 4.1. System Model

In an integrated RFID system such as Figure 1, a mass of tags belonging to multiple categories are distributed in the monitoring area. The tags have their respective IDs, and each ID is classified into two parts: category ID and member ID. Category ID is a binary string, and as an identifier, represents that a tag belongs to a certain category. In the binary string, one bit is 1, and all other bits are 0 s. Member ID indicates that a tag is a member of one particular category. Multiple readers are carefully deployed across the entire monitoring area to ensure communication between readers and tags.

The execution of MMTD is compatible with the ALOHA protocol, and we assume that all tag IDs are stored on a central server. Only one hash function is required for execution, and the data is stored in the tags’ memory. In one execution frame, the parameters including the frame size and a random hash seed are sent to tags by reader, once the tags receive the parameters, they will determine which slot to respond to. We classify the frame slots into two types: singleton slot and non-singleton slot. In singleton slot, only one tag maps into, and we represent it as 1; in non-singleton slot, there are no tag maps into or multiple tags map into, we represent it as 0.

To perform a fair simulation, we set the parameters as follows: The transmission rate of down-link is 53 Kb/s, while the transmission rate of up-link is 26.5 Kb/s. The duration time between the execution of two slots $\tau_{w}$ is 302 ms [6,24]. Following the assumptions made in previous missing tag detection schemes [3], we assume that there are no errors during wireless transmitting–receiving. We implemented our simulator using MATLAB.

### 4.2. Problem Definition

In an integrated RFID system, we assume that *U* readers are carefully deployed in the monitoring area to ensure that each tag can communicate with at least one reader. *N* tags are expected to be located in the monitoring area and distributed into λ categories $C_{1},C_{2},\ldots ,C_{\lambda }$. The number of tags in each category is $n_{i}$, i∈[1,λ]. In actuality, there are $M_{i}$ missing tags in the *i*-th category. For the *i*-th category, the number of present tags is $n_{i}-m_{i}$.

We use $t_{i}$ to represent the threshold value of expected missing tags. Let $p_{i}$ be the probability that the readers can detect the missing event of the *i*-th category. The definition of missing tag detection is demonstrated as follows.

**Definition** **1.**
*For the i-th category, the number of the expected present tags is $n_{i}$, the population size of missing tags is $m_{i}$, the optimum missing tag detection is at least one missing tag is found in a minimum execution time with the detection probability $p_{i}\geq \alpha_{i}$, if $m_{i}\geq t_{i}$, where $\alpha_{i}$ is the probability of detection requirement of the i-th category.*


However, in an integrated RFID system, *U* readers are deployed to monitor the tags which are attached to items. Thus, the tags outside the monitoring area of one reader will be determined as missing tags. However, the determined missing tags by one reader are not true missing tags.

**Definition** **2.**
*(The true missing tags). U readers are deployed with overlaps in an integrated RFID system. For the i-th category, after multiple turns of detection, each reader obtains a set of missing tags that are determined by itself. The true missing tags set $S=S_{1}⋂S_{2}⋂\ldots ⋂S_{j}⋂S_{U}$, where $S_{j}$ is the missing tags set which are obtained by the j-th reader.*


Thus, if the set *S* is empty, all expected tags present in the monitoring RFID system, whereas if there is at least one tag in the set *S*, the missing tag event is detected. As shown in Figure 1, for category one, the reader R1 obtains a missing tag set $S_{1}$ which contains tag3, tag2, tag4, tag5. $S_{2}$ contains tag1, tag3, tag4, tag5. $S_{3}$ contains tag1, tag3, tag2. $S_{4}$ considers all tags in category one are missing. The intersection of missing tag set *S* contains only one missing tag tag3. The tag3 is truly missing. The main notations used in this paper are shown in Table 1.

## 5. The Proposed MMTD Protocol

### 5.1. Protocol Design

We do not know whether all tags can exchange information with readers. In MMTD, the operations that are executed on each reader are similar to those in the simultaneous missing tag detection plus category clustering (SMTD + CC) protocol [6]. The SMTD+CC protocol is disembodied because it simply assumes that there is only one reader in an RFID system.

As shown in Figure 2, based on Manchester encoding, the combined signal received by the reader can be decoded into λ category vectors. We can obtain three category vectors from the actual time frame by decoding, which represent the mapping of each category. The number of tags for each category can be obtained through an estimation method in the monitoring area of one reader. For example, SEM [20] decodes the combined signal of multiple categories into categorized independence signals, and calculates the proportion of empty slots to estimate the population size of each category. The missing event will be issued at the end of the iterative operation by comparing the category vectors and the expected time frame. For example, the first slot should be 1 according to the expected time frame, but in category-1 vector, it is 0 in actuality, so we can determine that this tag is missing.

The traditional missing tag detection method is category by category for an integrated RFID system; this method is low time-efficient. MMTD designs a characteristic tag ID and adopts the parallel detection method to improve the time-efficiency. However, the setting of broadcast frame size is a problem that directly affects the detection efficiency. As known to us, in a single category RFID system, the detection efficiency is highest when the broadcast frame size is equal to the population size of tags. However, there are different population sizes in different categories within an integrated RFID system. They adopt the same broadcast frame size, more collision slots will present if the population size of a category is far greater than the broadcast frame size, whereas more empty slots will present if the population size of the category is far less than the broadcast frame size. These non-optimal solutions result in low utilization of slots in the missing tag detection. Thus, MMTD groups the similar population sizes of categories into one batch and sets the same parameters to the members of the batch. The parameters include the broadcast frame size and the random seed.

In order to obtain a higher time-efficiency, MMTD clusters the categories into multiple groups. We name this method for CG. CG is a *k*-means clustering algorithm, *k* is the number of clustering groups, means is the average. The cluster objects are $\left\{\langle n_{r1},f_{r1}\rangle ,\langle n_{r2},f_{r2}\rangle ,\ldots \langle n_{ri},f_{ri}\rangle ,\langle n_{Ui},f_{Ui}\rangle \right\}$, where r∈[1,U]. As shown in Figure 3, we assume that there are 10 categories monitored by one reader, and the number of tags in each category are 800, 900, 1200, 1400, 1550, 2200, 2400, 2550, 2800, 3000, respectively. The broadcast frame size in one batch is the average population size of tags in one category.

For example, if the *k* is 3, the categories will be clustered into 3 batches: $\left\{C_{0},C_{1},C_{2},C_{3},C_{4}\right\}$, $\left\{C_{5},C_{6},C_{7}\right\}$, and $\left\{C_{8},C_{9}\right\}$. Their broadcast frame sizes are 1170, 2383, 2900, respectively. However, we do not know the number of tags in each category under the monitoring of every reader. Thus, the optimal *k* is a variable for each reader. Before MMTD execution, the optimal *k* value will be obtained by simulation.

Specifically, in MMTD detection, each tag also selects a pseudo-random time slot to send its category identifier (category ID), and each reader can decode a frame occupation vector for each category. For each tag in category $C_{i}$, on reader $R_{r}$, we use the hash parameters and its ID to calculate the location in a given time frame. If the corresponding bit in the frame occupation vector is 0, we know that this tag is missing; it will be added into the missing tag set $S_{ri}$, which is initialized as an *∅*. After several rounds of tag detection operations, the set $S_{ri}$ may contain the actual missing tags and the tags that are covered by other readers. To obtain the real missing tag set $S_{i}$ for category $C_{i}$, we calculate $S_{i}=\cap_{r=1}^{U}S_{ri}$. This means that a tag is identified as missing if and only if it is recognized as missing by all readers. In this case, the missing tag event for category $C_{i}$ is found, where i∈[1,λ]. To meet the requirement of detection accuracy for each category, we must perform a sufficient number of detection rounds on each reader $R_{r}$. The detection rounds required by category $C_{i}$ on reader $R_{r}$ are represented by $X_{ri}$.

### 5.2. Parameters Optimization

We first investigate what is required by each reader $R_{r}$ to execute a number of frames (denoted as $X_{ri}$) to meet the requirement of detection accuracy for each category. Then, we calculate the optimal frame sizes $f_{ri}^{op}$ and ${f^{\prime }}_{ri}^{op}$, which minimize the time cost on reader $R_{r}$ for category $C_{i}$. We use $n_{ri}$ to represent the number of tags in category $C_{i}$ that are covered by the reader $R_{r}$.

#### 5.2.1. Number of Frames $X_{ri}$

We use $p_{ri}^{0}$ to represent the probability that an arbitrary bit in the frame occupation vector of category $C_{i}$ on reader $R_{r}$ is 0. The expression of $p_{ri}^{0}$ is calculated as follows.
(1)$$\begin{array}{c} \;p_{ri}^{0}={\left(1-\frac{1}{f_{ri}}\right)}^{n_{ri}}, \end{array}$$
where $f_{ri}$ is the broadcast frame size. However, for some practical reason [24], the frame has to terminate after executing ${f^{\prime }}_{ri}$ time slots, where ${f^{\prime }}_{ri}\;=\;\min\left\{f_{ri},512\right\}$ [25,26,27] is called the executed frame size. After executing $X_{ri}$ frames with sizes $f_{ri}$ and $f_{ri}^{\prime }$, the probability that a real-missing tag in category $C_{i}$ can be detected as absent from reader $R_{r}$ at least once is as follows: (2)$$\begin{array}{c} 1-{\left(1-\frac{f_{ri}^{\prime }}{f_{ri}}\times p_{ri}^{0}\right)}^{X_{ri}}. \end{array}$$

Hence, we can calculate the probability of a real-missing tag can be discovered as follows: (3)$$\begin{array}{c} {\left[1-{\left(1-\frac{f_{ri}^{\prime }}{f_{ri}}\times p_{ri}^{0}\right)}^{X_{ri}}\right]}^{U}. \end{array}$$

If $M_{i}$ tags in category $C_{i}$ are missing from the system ($M_{i}=t_{i}$), the missing tag event can be discovered if and only if at least one of them can be discovered. The corresponding probability is calculated as follows: (4)$$\begin{array}{c} 1-{\left\{1-{\left[1-{\left(1-\frac{f_{ri}^{\prime }}{f_{ri}}\times p_{ri}^{0}\right)}^{X_{ri}}\right]}^{U}\right\}}^{t_{i}}. \end{array}$$

To satisfy the detection probability $\alpha_{i}$, we must solve the following inequality,
(5)$$\begin{array}{c} 1-{\left\{1-{\left[1-{\left(1-\frac{f_{ri}^{\prime }}{f_{ri}}\times p_{ri}^{0}\right)}^{X_{ri}}\right]}^{U}\right\}}^{t_{i}}\geq \alpha_{i}. \end{array}$$

Solving the above inequality, we have: (6)$$\begin{array}{c} X_{ri}\geq \frac{\ln\left(1-\sqrt[{U}]{1-\sqrt[{t_{i}}]{1-\alpha_{i}}}\right)}{\ln\left[1-\frac{{f^{\prime }}_{ri}}{f_{ri}}{\left(1-\frac{1}{f_{ri}}\right)}^{n_{ri}}\right]}. \end{array}$$

To save time, we set $X_{ri}$ as follows: (7)$$\begin{array}{c} X_{ri}=\frac{\ln\left(1-\sqrt[{U}]{1-\sqrt[{t_{i}}]{1-\alpha_{i}}}\right)}{\ln\left[1-\frac{{f^{\prime }}_{ri}}{f_{ri}}{\left(1-\frac{1}{f_{ri}}\right)}^{n_{ri}}\right]}. \end{array}$$

#### 5.2.2. Optimize Frame Size $f_{ri}$ and ${f^{\prime }}_{ri}$

The time cost required by reader $R_{r}$ to satisfy the accuracy of category $C_{ri}$, denoted as $T_{ri}$, could be calculated as follows:(8)$$\begin{array}{cc} \;T_{ri} & =X_{ri}\times \left(\tau_{ip}+{f^{\prime }}_{ri}\times \tau_{ci}\right)\\ \; & =\frac{\ln\left(1-\sqrt[{U}]{1-\sqrt[{t_{i}}]{1-\alpha_{i}}}\right)}{\ln\left[1-\frac{{f^{\prime }}_{ri}}{f_{ri}}{\left(1-\frac{1}{f_{ri}}\right)}^{n_{ri}}\right]}\times \left(\tau_{ip}+{f^{\prime }}_{ri}\times \tau_{ci}\right), \end{array}$$

$\tau_{ip}$ on behalf of the slot duration for the reader initializes a $f_{ri}$-slot time frame with query command, and $\tau_{ci}$ represents the slot duration for the tag replies to the query command. Then, we provide sufficient analysis of the frame size of broadcast $f_{ri}$, and the frame size of execution $f_{ri}^{\prime }$ to make an impact on the time-efficiency of MMTD and propose Theorem 1 to recommend it in the following content.

**Theorem** **1.**
*For an arbitrary reader $R_{r}$, we assume the number of tags in each category $C_{i}$ covered by it, denoted as $n_{ri}$, is known to us. To minimize the detection time $T_{ri}$, the optimal broadcast frame size should be set to $f_{ri}^{op}\;=\;n_{ri}+1$, the optimal executed frame size ${f^{\prime }}_{ri}^{op}$ is $\min\{f_{ri}^{op},512\}$, i∈[1,λ] and r∈[1,U].*


**Proof.** 

(9)
$$\begin{array}{cc} \;\frac{\partial T_{ri}}{\partial f_{ri}} & =\frac{{f^{\prime }}_{ri}{\left(1-\frac{1}{f_{ri}}\right)}^{n_{ri}}}{\ln^{2}\left(1-\frac{{f^{\prime }}_{ri}}{f_{ri}}{\left(1-\frac{1}{f_{ri}}\right)}^{n_{ri}}\right)}\\ \; & \times \frac{\ln\left(1-p_{ri}\right)\left(n_{ri}+1-f_{ri}\right)\left(\tau_{ip}+{f^{\prime }}_{ri}\times \tau_{ci}\right)}{\left[1-\frac{{f^{\prime }}_{ri}}{f_{ri}}{\left(1-\frac{1}{f_{ri}}\right)}^{n_{ri}}\right]}. \end{array}$$

We investigate from Equation (Equation 9) that $\frac{\partial T_{ri}}{\partial f_{ri}}=0$ when $f_{ri}=n_{ri}+1$; $\frac{\partial T_{ri}}{\partial f_{ri}}<0$ when $f_{ri}<n_{ri}+1$; $\frac{\partial T_{ri}}{\partial f_{ri}}>0$ when $f_{ri}>n_{ri}+1$. We discover that the $T_{ri}$ achieves its minimum value when $f_{ri}^{op}$ is $n_{ri}+1$. Next, in order to optimize the ${f^{\prime }}_{ri}$, we obtain the first-order partial derivative against $T_{ri}$ as follows:
(10)$$\begin{array}{c} \;\frac{\partial T_{ri}}{\partial {f^{\prime }}_{ri}}=\frac{B\ln(1-p_{ri})}{\ln^{2}\left[1-\frac{{f^{\prime }}_{ri}}{f_{ri}}{\left(1-\frac{1}{f_{ri}}\right)}^{n_{ri}}\right]}, \end{array}$$
where $B=\tau_{ci}\ln(1-\frac{{f^{\prime }}_{ri}}{f_{ri}e})+\frac{\tau_{ip+f_{ri}^{\prime }\tau_{ci}}}{f_{ri}e-{f^{\prime }}_{ri}}$. Then, we obtain the first-order partial derivative of B with respect to $f_{ri}$ as follows:

(11)
$$\begin{array}{c} \;\frac{\partial B}{\partial f_{ri}}=\frac{-{f^{\prime }}_{ri}^{2}\tau_{ci}-f_{ri}e\tau_{ip}}{{\left(f_{ri}e-{f^{\prime }}_{ri}\right)}^{2}f_{ri}}<0. \end{array}$$

Note that when $f_{ri}$ takes its optimal value of $\hat{n_{ri}}+1$, the B is larger than 0. $\frac{\partial T_{ri}}{\partial {f^{\prime }}_{ri}}$ is smaller than 0, so the detection time $T_{ri}$ is a reduction function with respect to $f_{ri}^{\prime }$. Hence, the ${f^{\prime }}_{ri}^{op}$ should be set to $\min\{f_{ri}^{op},512\}$ for category $C_{i}$ on reader $R_{r}$.    □

The number of tags in different tag categories covered by the same reader are usually different. Thus, different tag categories need to be set differently to optimal frame sizes. Similar to the CC scheme [6], we divide tag categories under a reader into multiple batches according to the required frame sizes. The categories within the same batch share the same average frame size. The best batch division result is that the sum of time cost consumed by all batches is minimized. Note that the batch division process is virtually computed on the server side. After getting the best batch division strategy, missing tag detection is performed batch by batch.

#### 5.2.3. Dynamically Adjusting and Determining Termination

According to Theorem 1, when $f_{ri}\;=\;n_{ri}+1$, and ${f^{\prime }}_{ri}\;=\;\min\left\{f_{ri},512\right\}$, the detection time of category $C_{i}$ on reader $R_{r}$ is minimized. However, as aforementioned, the value of $n_{ri}$ is actually unknown to us. Hence, we will discuss how to configure the frame sizes of MMTD without knowing the value of $n_{ri}$. At the very beginning, we know nothing about $n_{ri}$, hence, we simply set $f_{ri}=n_{i}$ and $f_{ri}^{\prime }=\min\left\{512,f_{ri}\right\}$, where $n_{i}$ is equal to the population size of tags of category $C_{i}$ in the whole system. Then, we quote the CC protocol [6] to cluster the tag categories into multiple batches. Here, in each batch, the actually used frame sizes are averaged from the optimal frame sizes of all categories in this batch. After executing the first round of missing tag detection for each batch on reader $R_{r}$, the obtained frame occupation vectors can also be fed into the SEM algorithm [20] to estimate the value of $n_{ri}$ for each category $C_{i}$. For the following rounds of detection, we can use the approximate value $n_{ri}$ estimated from the previous round to optimize the frame sizes and adjust the batch division.

On a reader $R_{r}$, a category may belong to different batches in different rounds because the estimated tag cardinality $n_{ri}$ is not entirely accurate. Thus, a category $C_{i}$ may exploit different frame sizes in different rounds. In addition, it is hard to simply calculate how many rounds are required to satisfy its detection probability. Next, we propose a dynamic determination method that allows us to judge whether a category has met the required detection accuracy at the end of an arbitrary round. Suppose that the reader $R_{r}$ has performed *y* rounds of detection for category $C_{i}$. In the *x*-round (x∈[1,y]), for each category $C_{i}$, we obviously know the ratio $p_{x}^{0}$ of bit 0 s in the corresponding frame occupation vector and the actually used frame sizes $f_{ri}^{x}$ and $f_{ri}^{\prime x}$. Thus, we can easily calculate the probability that a real-missing tag can be detected out in this round as $p_{x}^{0}\times \frac{f_{ri}^{\prime x}}{f_{ri}^{x}}$. The overall probability that a real-missing tag can be detected out through *y* rounds can be easily given as $1-\prod_{x=1}^{y}\left(1-p_{x}^{0}\times \frac{f_{ri}^{\prime x}}{f_{ri}^{x}}\right)$. If this probability is larger than $\sqrt[{U}]{1-\sqrt[{t_{i}}]{1-\alpha_{i}}}$, the detection process for category $C_{i}$ on reader $R_{r}$ can be terminated due to the following reasons. Each reader has the probability $\sqrt[{U}]{1-\sqrt[{t_{i}}]{1-\alpha_{i}}}$ to detect a real-missing tag. All *U* readers will have the probability $1-\sqrt[{t_{i}}]{1-\alpha_{i}}$ to detect a certain real-missing tag, it is eventually asserted as a missing tag. At least one of the $t_{i}$ real-missing tags is detected out, and we will successfully assert the missing event for category $C_{i}$. Hence, the probability of successful detection for category $C_{i}$ can be calculated as $1-{\left[1-\left(1-\sqrt[{t_{i}}]{1-\alpha_{i}}\right)\right]}^{t_{i}}$, which exactly equals the required probability $\alpha_{i}$. This means that the above dynamic termination approach can satisfy the predefined missing tag detection accuracy for each category.

### 5.3. The Attacks from Physical Layer

As we know, RFID tags are commonly exposed to the outside of the tagged items, and the communication between reader and tags through electromagnetic waves. Thus, the tags are easy to destroy with physical attacks, and the tag identification is vulnerable to electromagnetic interference attacks. These attacks on the physical layer take advantage of the nature of RFID communication, the poor physical securing of RFID devices, and the lack of tensile strain capacities against physical damage. The attacks cause three types damage to RFID devices: permanently disabling damage, temporary damage, and persistent damage. The permanently disabling damage to RFID devices is implemented by tag removal, destructing, and the *Kill* command issue. The temporary damage adopts the signal shielding method to escape the tag identification. For example, an aluminum bag prevents the communication between reader and tags. The persistent damage is that the adversarial devices are deployed surreptitiously to replace the legitimate RFID reader and tags.

The first two damages can be detected by missing tag detection protocols and artificial reinspection, the killed tags can be recovered by the *activate* command. However, this persistent damage is hard to detect by the common missing tag detection protocols. Many related protocols concentrated on this damage, they adopted encryption methods to prevent communication between adversarial devices and legitimate devices. For example, Qi et al. [28] proposed an encryption method that overcomes the inefficiency hurdles of CP-ABE [29]. Qi et al. adopted a double encryption paradigm to prevent the malicious intrusion reader and tag information loss with high efficiency. Gope et al. [30] proposed a enhanced protocol based on physically unclonable functions to prevent invasion readers. Though physical security is a concern among researchers, many excellent works have been proposed, and this paper concentrates on the time-efficiency of missing tag detection. Thus, we do not discuss more about physical security information in RFID systems.

### 5.4. The Detection of R2Rc

R2Rc is reader-to-reader collision, that is to say, two readers communicate with one tag at the same time. The tag receives a combined signal from readers, and the reply from the tag is an error information. The two readers can not decode the tag ID from the error information. Thus, R2Rc is a problem that cannot be ignored in RFID research. How do we determine whether R2Rc exists between two readers? An intuitive approach is tag identification, if two readers obtain the same tag ID, we can determine that the R2Rc exists. The other method is to obtain the Received Signal Strength Indicator (RSSI) from one location to ensure the communication border of one reader. If the communication borders of readers intersect, we can determine that R2Rc exists.

After confirming the relationship between readers and whether the R2Rc exists in any two readers, the next question is how to eliminate the influence of R2Rc. Most existing protocols are designed based on Time Diverse Multiple Access (TDMA), all readers are split into multiple groups, there is no R2Rc in each group, and the groups work at different time frames. For example, the reader scheduling protocol Colorwave [31] groups the readers into many colored nodes, the nodes of the same color can work simultaneously. However, in multiple categories RFID system, they still detect the missing tags category-by-category. This method is low time-efficiency. To minimum the impact of R2Rc, some protocols require that the readers work on different frequency communication channels. However, the communication range of the reader is closely related to the working frequency. The higher the working frequency, the longer the communication distance. Thus, changing the frequency communication channel may result in tag identification failure.

## 6. Performance Evaluation

In this section, we evaluate the performance of MMTD in two key aspects: detection accuracy and time efficiency. To demonstrate the outstanding performance of the proposed MMTD, we import multiple comparison protocols. The comparison protocols include TRP [7], RUN [3], SSDA, ESSDA [32], EMD [8], and SMTD, SMTD+CC [6]. The slotted ALOHA scheme is their fundamental method, TRP computes the probability of one missing tag mapping into an empty slot; unexpected tags are the impact factor that decreases the time-efficiency for missing tag detection, RUN can accomplish the detection effort with a high time-efficiency in this environment. SSDA and ESSDA detect a transformation that the expected singleton slot is changed into an actual empty slot, to determine whether the mapping tag is missing. However, the utilization of slots is low, because the collision slots are abandoned directly. EMD adopts a higher time-efficient sampling method to reduce the number of response tags; SMTD and SMTD+CC can simultaneously detect missing tags for multi-category through combine-signal decoding.

MMTD is workable in various static tagged applications that have different spatial limitations. In this paper, to give an intuitive demonstration and comparable simulation, an integrated RFID system is built, we set 20 readers monitor a 60 m × 60 m monitoring area. According to the reader scheduling protocols [31], the 20 readers are split into three groups, and they are marked with cyan, red, and black. The locations of readers marked with cyan are (46.8,55.8), (1.2,43.2), (25.2,34.8), (53.4,36.0), (9.0,6.6), and (38.4,5.4), respectively. Their communication radii are 9.0 m, 12.0 m, 15.6 m, 11.4 m, 12.0 m, and 10.8 m, respectively. The locations of readers marked with red are (1.8,58.2), (35.4,53.4), (57.6,52.8), (16.2,46.2), (10.2,28.2), (41.4,19.8), (17.4,11.4), and (58.2,4.2), respectively. The corresponding communication radii are 9.0 m, 9.0 m, 9.0 m, 10.2 m, 7.8 m, 13.2 m, 9.6 m, and 8.4 m, respectively. The locations of black readers are (18.0,57.0), (46.2,42.0), (4.8,22.2), (55.2,22.2), (27.6,7.8), and (49.2,5.4), respectively. Their communication radii are 14.4 m, 12.0 m, 12.0 m, 12.0 m, 12.0 m, and 12.0 m, respectively. The distribution of readers as shown in Figure 4. The detection time begins when the first query starts, until the last reader finishes the missing tag detection task.

In the simulation, we verify that the detection accuracy of our MMTD meets the requirements, and that the execution time of MMTD is shorter than the comparison protocols. We assume that there are 20 readers randomly deployed in the monitoring area to monitor 10 categories of tags. The sizes of category-1 to category-10 are 1500, 2000, 3000, 3500, 8000, 9500, 20,000, 21,000, 22,000, and 23,000, respectively [6]. All of the tags are randomly distributed into the monitoring area, and each tag can be monitored by at least one reader. The number of missing tags $M_{i}$ in each category is set to 10. In the default setting of MMTD, the threshold $t_{i}$ on the number of missing tags is 10 and the probability of missing tags detection $\alpha_{i}$ is 0.95. Since the number of missing tags in every categories reaches the threshold, MMTD should theoretically detect out at least one missing tag in each categories with 0.95 probability.

### 6.1. Detection Accuracy

To evaluate the detection accuracy of the proposed MMTD protocol, we deploy 20 readers in the monitoring area. The true number of missing tags varied from 8 to 12. We observe in Figure 5a–e that the probability of detection satisfies the requirements when the population size of missing tags equals or exceeds the tolerance threshold of 10 in each category; therefore, the MMTD satisfies the accuracy requirements in multi-category and multi-reader RFID scenarios.

However, when the tolerance threshold is larger than the number of missing tags, the probability of detection does not always satisfy the required 0.95. This is because the actual execution rounds are smaller than the requirements, and this happens when the population size of the category is larger than the broadcast frame size in its batch.

### 6.2. Time-Efficiency in Multi-Reader Scenarios

In an actual RFID scenario, the distribution area of one category tags may be monitored by multiple readers. Though the burden of each reader is reduced and the size of each frame is decreased, the number of detection frames is increased because each reader needs to satisfy stricter detection accuracy requirements. In this section, we investigate the time-efficiency of MMTD when the various parameters are changed.

#### 6.2.1. Impact of *U*

The value of *U* is the number of readers required to be deployed into the monitoring area. We enumerate the *U* data from 20 to 40. At the moment, the other parameters are the default setting, the μ and σ are not workable, because the the population sizes of 10 categories are constant. Two major observations are shown in Figure 6. First, the proposed MMTD protocol is the fastest protocol for changing the number of the readers. For example, the execution time of SMTD+CC is 52.9 s, however, the time required by the MMTD is only 17.9 s when there are 40 readers in the monitoring area. This result indicates that the MMTD runs 8.0× faster than the RUN protocol. Second, the time consumed by the MMTD reduces as *R* increases because the burden of the reader decreases. However, the time consumed by other related works is rising, because an increasing number of readers will bring more groups into reader-scheduling.

#### 6.2.2. Impact of μ

The average value μ in the normal distribution function 〈μ,σ〉 determines the number of tags directly. In this simulation, we vary the value of μ from 10,000 to 20,000. That is to say, the number of tags is monotonically increasing. The value of *R* is 20, and the other parameters are the same as those in the part of “Impact of *U*”. According to the demonstration of Figure 7, the MMTD protocol has the highest time-efficiency among the comparison protocols. For example, if the data of μ is 20,000, the time consumption of SMTD + CC is 47.4 s; however, the execution time of the MMTD protocol is only 27.0 s. Moreover, as the μ increases in each category, the rate of change of the MMTD is much lower than that of the other protocols.

#### 6.2.3. Impact of σ

With the standard deviation σ of the function increasing which determines the number of tags, the volatility of the number of tags is monotonically increasing. To make the comparison under different σ, we vary the value from 5000 to 10,000 in each category in the simulation. The MMTD protocol has the best performance by changing σ as shown in Figure 8. For example, if the σ is 10,000, the execution time of SMTD + CC is 50.4 s. However, that of the MMTD is only 28.4 s. This means that our MMTD runs 1.7× faster than the SMTD+CC protocol. As σ increases, the MMTD protocol performs much better than other comparison protocols. The reason is that the broadcast frame size of the MMTD is smaller than that of the SMTD + CC, and the ratio of the detection time of the MMTD to the number of broadcast frame size of the MMTD is not positive.

#### 6.2.4. Impact of $t_{i}$

The tolerance threshold $t_{i}$ determines the performance of MMTD directly. The smaller $t_{i}$ means the stricter requirement on detection accuracy. To show the excellent performance of MMTD, we make the comparison between the protocols when we vary the tolerance threshold $t_{i}$ from 10 to 20 for each category. The number of actual missing tags $M_{i}$ is equal to $t_{i}$. We observe the following in Figure 9. First, the MMTD has the best performance of the protocols. For example, if $t_{i}=10$, the execution time of SMTD + CC is 53.0 s, whereas that of the MMTD protocol is only 30.2 s. This means that our MMTD runs 1.75× faster than the SMTD + CC protocol. Second, as $t_{i}$ increases, the time consumption of TRP, EMD, RUN, and SMTD + CC shows a decreasing trend, but this trend is not obvious in terms of MMTD. The reason is that the execution time of all protocols are inversely proportional to the tolerance threshold $t_{i}$ except for MMTD. The time-cost of the MMTD protocol is only slightly affected by the threshold $t_{i}$, as shown in Equation (Equation 8).

#### 6.2.5. Impact of $\alpha_{i}$

To clarify the comparison, we vary the reliability probability $\alpha_{i}$ from 0.9 to 0.99 in the simulation for each category. The following two observations are made in Figure 10. First, the proposed MMTD protocol exhibits the highest-efficiency among the comparison protocols. For example, when $\alpha_{i}$ is 0.99, the time-execution of SMTD + CC is 92.9 s, whereas that of the MMTD is only 37.7 s. This means that the execution speed of our MMTD runs 2.4× faster than the SMTD+CC protocol. Second, when the required detection accuracy $\alpha_{i}$ is higher, the time-cost of all protocols, except for ESSDA and MMTD increases significantly. The reason is that the detection times of the MMTD and ESSDA protocols are slightly affected by $\alpha_{i}$.

## 7. Conclusions

In this paper, we propose a highly efficient method for missing tag detection, which is designed for multi-category and multi-reader RFID scenarios. We make the following key contributions. First, MMTD optimizes the parameters of each reader to personalize the reader’s settings according to the tag populations, while the other protocols often broadcast identical settings by all readers. Second, MMTD proposes a collision elimination algorithm to detect missing tags simultaneously without any conflicts between multiple readers. Third, extensive simulations indicated that the MMTD protocol outperformed similar missing tag detection protocols by at least 1.5× in the multi-reader scenarios.

## Figures and Tables

**Figure 1 sensors-22-04601-f001:**
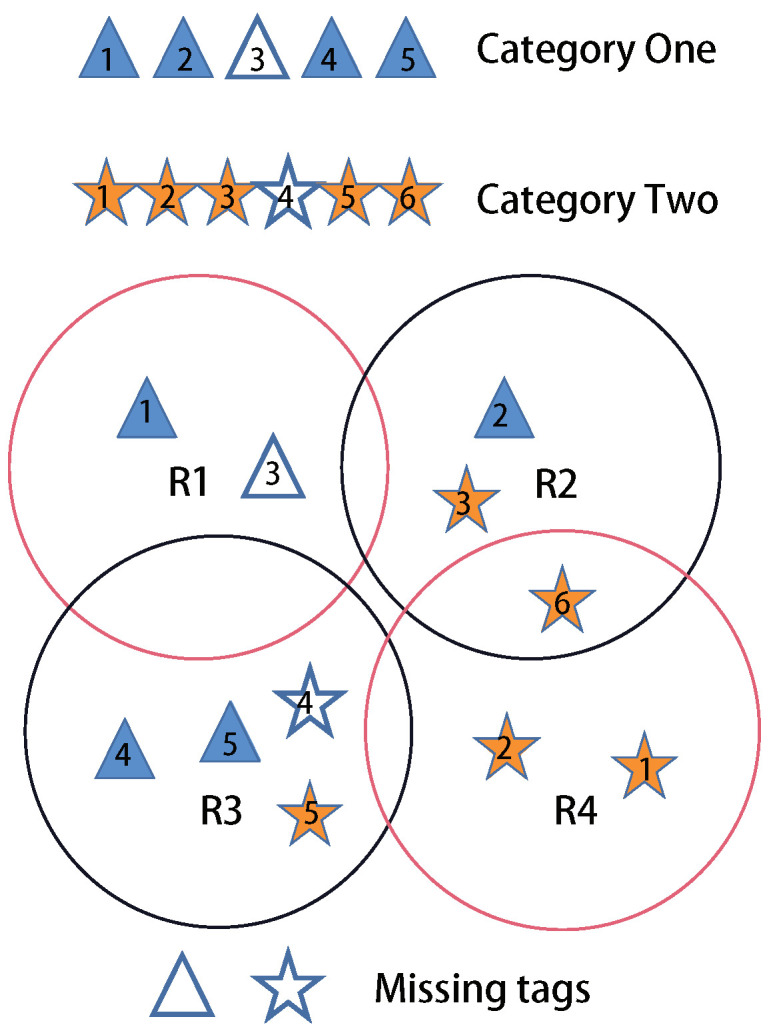
System model.

**Figure 2 sensors-22-04601-f002:**
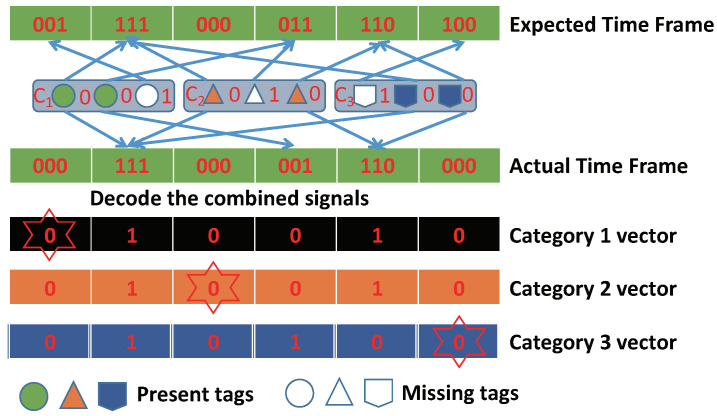
The decoding of received combined signal.

**Figure 3 sensors-22-04601-f003:**
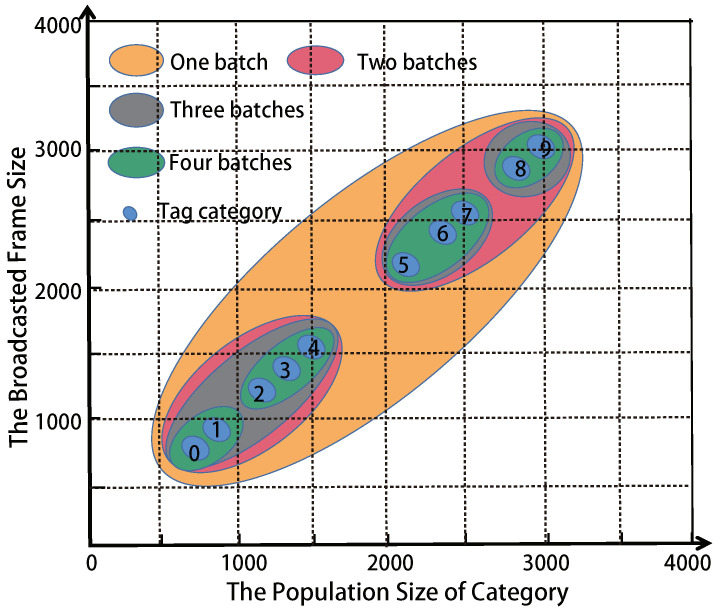
Category clustering.

**Figure 4 sensors-22-04601-f004:**
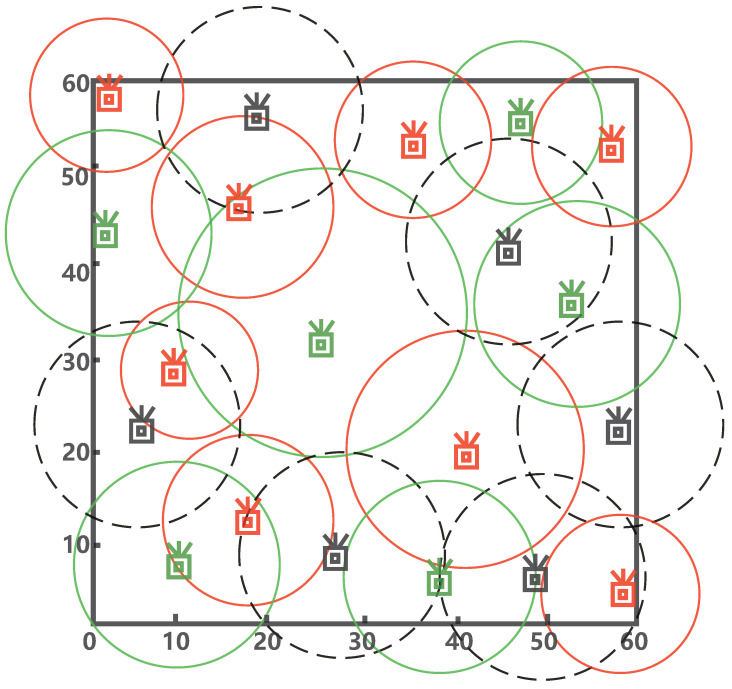
An example of reader distribution.

**Figure 5 sensors-22-04601-f005:**
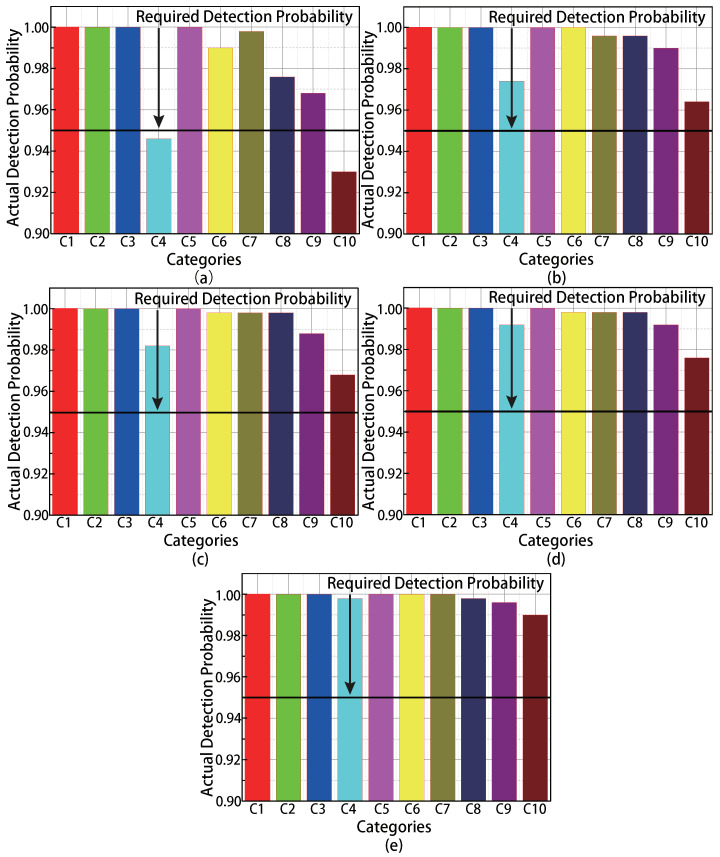
The relationship between detection accuracy and the number of actual missing tags. (**a**) 8 missing tags in each category; (**b**) 9 missing tags in each category; (**c**) 10 missing tags in each category; (**d**) 11 missing tags in each category; (**e**) 12 missing tags in each category.

**Figure 6 sensors-22-04601-f006:**
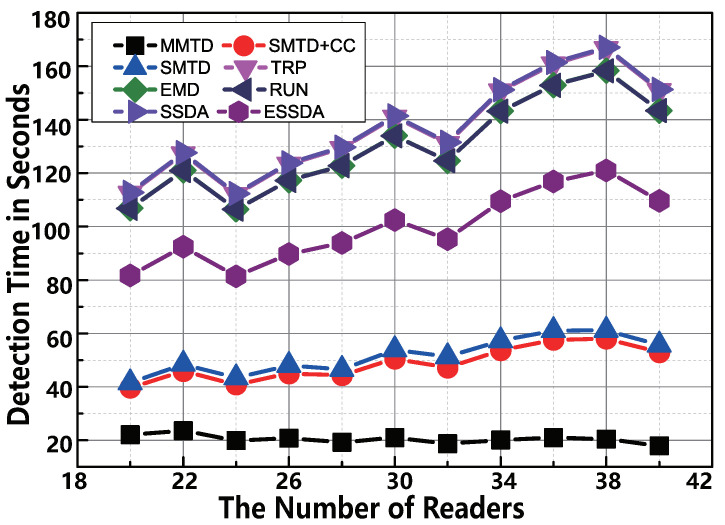
The comparison of time-efficiency between MMTD and other missing tag detection protocols when the number of readers *R* is set to be from 20 to 40.

**Figure 7 sensors-22-04601-f007:**
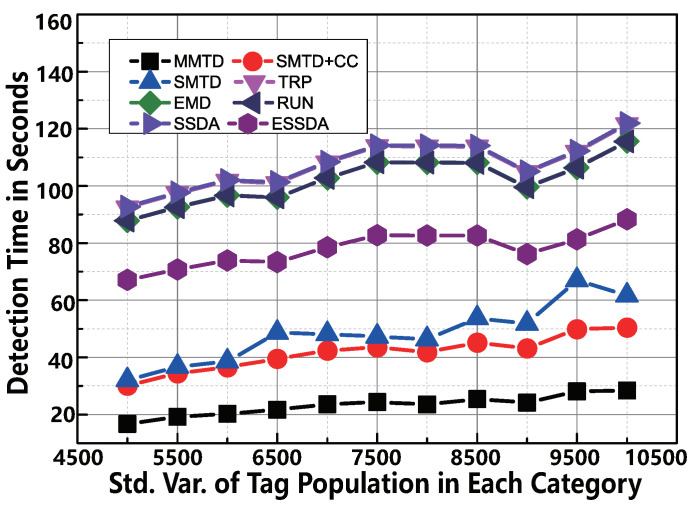
The comparison of time-efficiency between MMTD and other missing tag detection protocols when σ is set to be from 5000 to 10,000.

**Figure 8 sensors-22-04601-f008:**
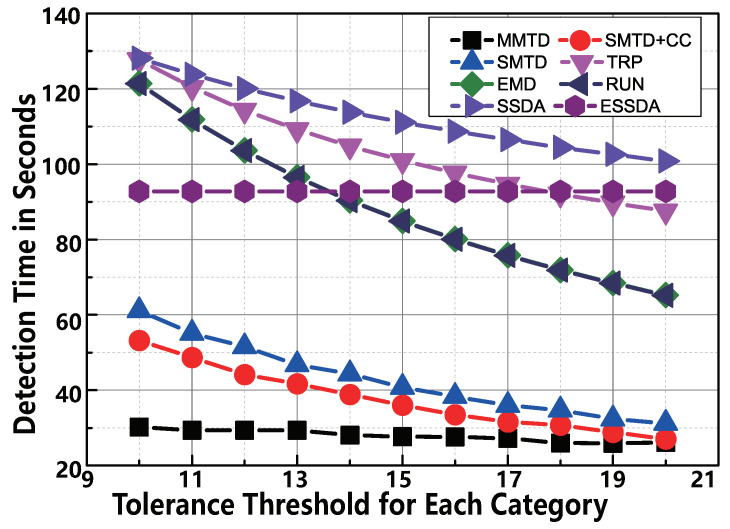
The comparison of time-efficiency between MMTD and other missing tag detection protocols when $t_{i}$ is set to be from 10 to 20.

**Figure 9 sensors-22-04601-f009:**
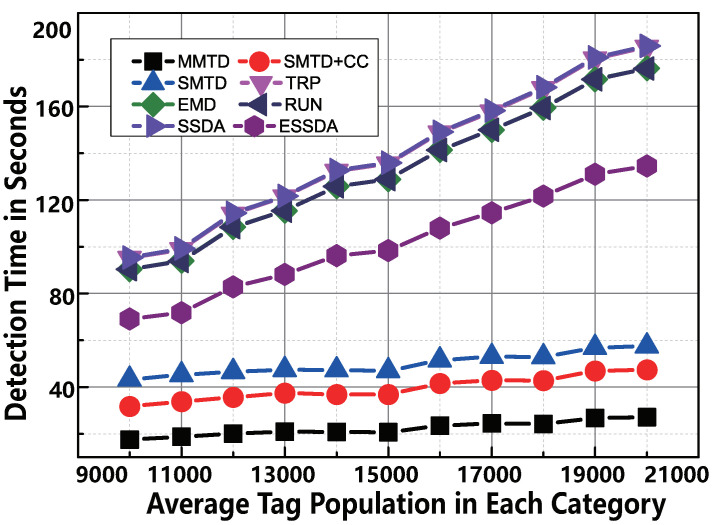
The comparison of time-efficiency between MMTD and other missing tag detection protocols when μ is set to be from 10,000 to 20,000.

**Figure 10 sensors-22-04601-f010:**
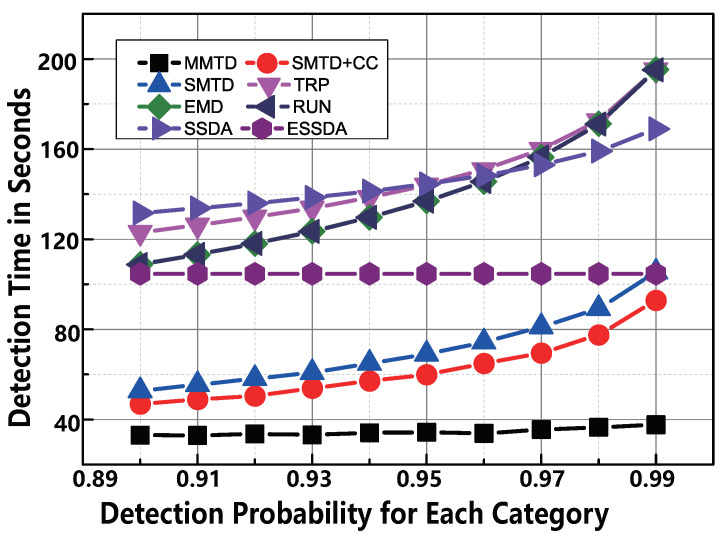
The comparison of time-efficiency between MMTD and other missing tag detection protocols when $\alpha_{i}$ is set to be from 0.9 to 0.99.

**Table 1 sensors-22-04601-t001:** Main notations used in the paper.

Notations	Descriptions
λ	number of categories in RFID system.
$C_{i}$	the *i*-th tag category.
*U*	number of readers.
$\alpha_{i}$	required detection probability of category
$t_{i}$	tolerant threshold of category $C_{i}$
$M_{i}$	number of actual missing tags in *i*-th category
$n_{i}$	number of known tags in category $C_{i}$
$M_{i}$	the finally missing tags set
$X_{ri}$	the detection rounds required by category $C_{i}$ on reader $R_{r}$
*s*	random hash seed for each round of frame.
$f_{ri}$	broadcast frame size for category $C_{i}$ on $R_{r}$.
$f_{ri}^{\prime }$	executed frame size for category $C_{i}$ on $R_{r}$.
$f_{ri}^{op}$	optimal broadcast frame size for $C_{ri}$.
${f^{\prime }}_{ri}^{op}$	optimal executed frame size for $C_{ri}$.
$T_{ri}$	detection time required by $C_{i}$.
μ	the average of the population size of tags in each category.
σ	standard variance of the population size of tags in each category.

## Data Availability

Not applicable.

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
