# Peer review of "A High Time-Efficient Missing Tag Detection Scheme for Integrated RFID Systems"

_sensors, 2022, doi:10.3390/s22124601_

Round 1
Reviewer 1 Report
This article discusses an interesting problem (missing tag problem). I have found following issues, which are required to be address.
- First Introduction part is too lengthy, first two paragraphs should be deleted;
- The formal analysis part is vague, more details are required.
- More experimental results are required to be added;
- The problem definition part needs to be more formal;
- Recently physical security of the Tag has become a real problem, authors need to consider that.
Reviewer 2 Report
The authors present a missing tag detection technique involving multiple readers. The models are clearly written and the experimental studies with good results have demonstrate the performance. My further concerns are summarized as follows.
- Since the authors claim the proposed method outperforms the state of the art, please give the competitive peers in a table with their key parameters including time-efficiency.
- A model of the missing tag detection technique is required to give the readers an intuitive understanding.
- The Language of this paper needs to be further polished. May presentations are incorrect.
Reviewer 3 Report
This work proposes a method for missing tag detection, targeting multi-category and multi-reader RFID scenarios. The proposed method optimizes the parameters of each reader to personalize the reader’s settings and proposes a collision elimination algorithm to detect missing tags simultaneously. The paper has well-defined the problem, developed the method, and conducted the simulation. But the authors shall include more descriptive explanations of graphs. For example, Figure 1 shows one example of reader deployment. But what does it mean? How to interpret it? Moreover, the authors are recommended to include a comparison table between their MMTD method and other methods, which can highlight the performance of their method.
Round 2
Reviewer 1 Report
I think the authors have miss understood my comments on physical security. They can refer to the following papers for that
https://ieeexplore.ieee.org/abstract/document/8353855
Reviewer 2 Report
The authors have addressed all my concerns. The paper can be accepted in the present form.
Author Response
Thank you very much for handling and accepting this submission.